# Reliability of the Feeling Scale and Felt Arousal Scale in Older Adults

**DOI:** 10.3390/ijerph22091317

**Published:** 2025-08-25

**Authors:** Victor Grillo Sobrinho, Walace Monteiro, Carlos Alberto Aiello Ribeiro, Mauro Lúcio Mazini Filho, Rodrigo Gomes de Souza Vale, Nádia Souza Lima da Silva

**Affiliations:** 1Graduate Program in Exercise and Sport Science, State University of Rio de Janeiro, Rio de Janeiro CEP 20550-900, Brazil; walacemonteiro@uol.com.br (W.M.); carlosaiello1991@hotmail.com (C.A.A.R.); rodrigovale@globo.com (R.G.d.S.V.); nadiaslimas@gmail.com (N.S.L.d.S.); 2Federal Institute of Education, Science and Technology, Cataguases Campus, Cataguases 36773-563, Brazil; mazinifilho@gmail.com

**Keywords:** affect, felt arousal, affective training, aging

## Abstract

Affective responses to physical exercise vary according to sensory impressions and can influence motor performance. To assess affective states in exercise contexts, scales such as the Felt Arousal Scale and the Feeling Scale are commonly used. Although widely applied, their psychometric properties have not yet been tested in older adults, which limits their use in this population. This test–retest study aimed to test the reliability of these scales in older adults of both sexes. The sample consisted of 80 volunteers (67 women) engaged in either resistance training (*n* = 37) or functional training (*n* = 43), with a mean age of 70 ± 9 years. Data collection occurred on two separate days, with a minimum interval of 15 days. The following instruments were applied: (a) Felt Arousal Scale and Feeling Scale at pre-, during (minutes 20 to 25), and post-training; (b) Borg Scale during and after training; and c) Mini-Mental State Examination during the first visit. The results showed strong to very strong correlations for both scales (Feeling Scale: ρ = 0.936 during, ρ = 0.840 post; Felt Arousal Scale: r = 0.967 during, r = 0.887 post), as well as good internal consistency (Feeling Scale: α = 0.887; Felt Arousal Scale: α = 0.860) and excellent temporal stability (Feeling Scale: ICC = 0.925; Felt Arousal Scale: ICC = 0.869). It is concluded that both instruments are reliable for measuring affect and arousal in older adults who engage in physical exercise.

## 1. Introduction

Affective responses to physical exercise are sensitive to an individual’s sensory impressions, which can impose limitations on the tasks to be performed [1]. For example, the literature shows that exercise performed in natural environments tends to induce more positive affect compared to those conducted in enclosed spaces [2,3]. There is also evidence that listening to music during physical activity can enhance positive feelings [4,5]. On the other hand, high-intensity exercise generally has a negative impact on mood [6].

Moreover, arousal is a key factor in exercise performance, as it can increase the individual’s motivation, thereby contributing to adherence to a regular exercise routine [7,8]. When an individual feels enthusiastic and energized during exercise, they are more likely to maintain consistency in their physical activity. Thus, a person may experience either an increase or decrease in affect and arousal during an exercise session, which may be perceived as either positive or negative [9,10]. Among older adults, this aspect is particularly relevant, as emotional state can be a decisive factor in maintaining or abandoning physical activity, which is essential for their health and well-being [7].

Given this context, it is important to understand how physical exercise influences the practitioner’s affective state. To assess it in sport and exercise settings, health professionals have access to a wide range of validated questionnaires and scales, many of which share similarities and distinctions, such as the Exercise-Induced Feeling Inventory [11], the Profile of Mood States [12], and the Positive and Negative Affect Schedule [13]. These instruments are used to assess self-perception, typically expressed through paper-based responses or interviewer-administered questions, and are therefore highly language-sensitive.

Two widely used and notable affective scales in exercise-related research and interventions are the Feeling Scale (FS) [14] and the Felt Arousal Scale (FAS) [15]. The FS was developed to measure pleasure or displeasure experienced during physical activity and is used to evaluate how a person feels emotionally in response to different intensities of exercise [8], whereas the FAS measures the level of emotional arousal or activation—an important component of affective responses that may vary independently from feelings of pleasure or displeasure [9]. Although these scales are well established in studies involving other populations, they have not yet had their reliability assessed for use with older adults, which limits their applicability.

The FS and the FAS are widely used in exercise and sport science due to their brevity, ease of application, and minimal disruption to the activity being performed [14,15]. These scales are particularly suitable for use during exercise sessions, as they require only a few seconds to complete and can be administered repeatedly without interfering with the flow of the session. The theoretical foundation of both scales lies in the Affective Circumplex Model, which conceptualizes affective states along two primary dimensions: valence (pleasure–displeasure) and arousal (low–high activation) [16]. The FS captures the valence dimension, indicating the degree of pleasure or displeasure felt at a given moment, while the FAS assesses arousal, reflecting the level of physiological and psychological activation. Together, these instruments provide a concise yet comprehensive snapshot of an individual’s affective state in response to physical exertion.

Importantly, the reliable assessment of emotional states through instruments such as the FS and FAS can contribute to the early detection of psychological distress, enabling the development of targeted preventive strategies particularly in vulnerable populations such as older adults [17]. By identifying changes in affective responses during exercise, health professionals may be better equipped to implement timely interventions aimed at promoting mental well-being and long-term adherence to physical activity [18].

However, it is important to consider that the variability and reliability of affective scales such as the FS and FAS can differ according to exercise intensity. Previous studies have shown that reliability may vary slightly across intensities, especially due to physiological and perceptual fluctuations associated with effort [19]. Additionally, greater variability in affective responses has been observed at intensities below the lactate threshold, likely due to the broader range of subjective interpretations in this zone [18]. Nonetheless, these fluctuations tend to remain limited, and both FS and FAS generally present acceptable psychometric stability within typical exercise protocols.

Based on this premise, the aim of the present test–retest study was to test the reliability of the FS and FAS, considering the specific characteristics of this population.

## 2. Materials and Methods

### 2.1. Sample

The sample size was estimated using G*POWER software (Version 3.1.9.7, Heinrich Heine University Düsseldorf, Düsseldorf, Germany) [17,20], based on a significance level of *p* < 0.05, statistical power of 0.80, and a large effect size (ƒ^2^ = 0.4) [21], resulting in an estimated sample of 79 participants. The following inclusion criteria were adopted: (a) older adults of both sexes, aged 65 years or older; (b) previous experience with the specific type of training to which they were assigned within the last three months (participants in the Functional Training group were required to have prior experience with Functional Training, while those in the Resistance Training group were required to have previous experience with Resistance Training); and (c) a minimum attendance rate of 70% in their respective training sessions during the three months preceding the start of the study. Exclusion criteria included the following: symptoms of depression; mental or cognitive impairments that could interfere with comprehension of the proposed scales; inconsistent responses to the scales during the pre-training assessment, potentially indicating a pre-existing affective state that could influence subsequent responses during or following the exercise session; failure to attend scheduled visits; and scoring below 24 points on the Mini-Mental State Examination.

All participants signed an informed consent form, and the study was approved by the institutional research ethics committee (CAE 73881123.6.0000.5259). The initial sample consisted of 91 volunteers (76 women and 15 men), of whom 41 participated in resistance training (RT) and 50 in functional training (FT), with a mean age of 70 ± 9 years. During data collection, 11 participants were excluded: 2 for not reaching the minimum score on the Mini-Mental scale; 4 for not signing the Informed Consent Form; and 5 for providing inconsistent responses on the FS or FAS scales during the pre-training assessment, as illustrated in Figure 1.

### 2.2. Experimental Design

Data collection was conducted over four visits to the training center. The first two visits were dedicated to administering 10RM tests for the selected exercises. In the final two sessions, participants were randomized into two groups according to the type of training performed—resistance training (RT) or functional training (FT)—which was applied twice, with a 15-day interval between sessions. This interval was necessary to ensure the stability of emotional responses, which may vary significantly due to daily fluctuations in factors such as mood, stress, and life events. According to Ekkekakis and Petruzzello [22], a 15-day interval between sessions helps to obtain more stable and representative measures of participants’ emotional states, minimizing the influence of daily fluctuations. Additionally, it reduces the likelihood that participants will remember their previous responses to the questionnaires, which could bias subsequent measurements.

The exercise modalities investigated were chosen because they represent distinct methodological approaches. RT is traditionally structured with a focus on individualized and controlled exercises, typically performed using specific equipment in a gym setting. In contrast, FT adopts a more dynamic and integrated approach, often conducted in group settings and emphasizing multi-joint movements that simulate daily life activities.

To determine the reliability of the FS and FAS scales in assessing affective responses, the following procedures were adopted: (a) Participants received instructions on how to use the Borg, FS, and FAS scales; (b) the Mini-Mental State Examination was administered to identify any indication of cognitive impairment or depression; (c) the FS and FAS scales were administered pre-training, at the midpoint of the exercise session (20–25 min), and immediately post-training; (d) the Borg scale was applied halfway through and at the end of the training session. During the second visit, the entire data collection sequence from the first session was repeated, except for the Mini-Mental test.

### 2.3. Instruments

Before administering the scales, participants received clear and objective explanations about how each scale functioned. The scales were presented visually, using images and simple descriptions, adjusted as needed to ensure participant comprehension. Administration was conducted verbally, allowing participants to provide their responses directly and with clarity.

The Brazilian versions of the FS and the FAS, previously translated and culturally adapted into Portuguese, were used in the present study. These versions have demonstrated good comprehension and suitability for the Brazilian population. Prior studies conducted in Brazil have reported their practical applicability during physical exercise sessions, including among older adults, highlighting their syntactic clarity and ease of use, particularly during resistance training [23].

Data on affective responses and mental state were collected before training (to assess mood prior to activity), during, and after the training sessions, in order to evaluate the reliability of the scales using the following instruments:(a)Felt Arousal Scale (FAS) is used to assess the level of perceived arousal. This scale measures the perception of activation on six levels, ranging from low arousal 1 to high arousal 6 [15];(b)Feeling Scale (FS) is used to assess the level of perceived affective valence. It captures emotional responses in terms of pleasure or displeasure experienced during physical activity. The scale consists of eleven levels that reflect positive or negative sensations [14].

The Borg Rating of Perceived Exertion (RPE) scale was used to assess subjective perception of effort during physical activity and to monitor exercise intensity during both training sessions for each modality [20]. This instrument was chosen due to its established relationship with physiological indicators (e.g., heart rate and oxygen consumption) used to quantify exercise intensity, ensuring that effort was comparable across both sessions and modalities [24].

### 2.4. Training Sessions

The sessions for the RT group (Resistance Training) consisted of exercises performed alternately by body segment, following this sequence: knee extension on machine, seated row, leg press, lateral shoulder raise with dumbbells, abductor machine, seated chest press on machine, adductor machine, bicep curls with dumbbells, leg curl on machine, and triceps pushdown (pulley). This order was chosen to avoid excessive fatigue during the session, which is recommended for older adults [25]. Participants performed two sets of 10–12 repetitions using loads equivalent to 70% of their 10RM, with 2 min rest intervals between sets and exercises. Each session lasted approximately 40 min.

Prior to the start of the training protocol, participants completed a 10RM test to determine the load for each exercise. Five exercises were tested per day, following the same sequence used in the training sessions. A 15 min rest interval was provided between each 10RM test. Since the participants were already familiar with the equipment and exercises, a maximum of three attempts was required to establish the appropriate 10RM load for each exercise.

The sessions for the FT group (Functional Training) included the following exercises, performed in the following sequence: bodyweight squat, one-arm row with dumbbells, forward lunge with movement, shoulder press with dumbbells, standing calf raise, high pull with kettlebell, marching in place with high knees, step-up, and zigzag walk with low obstacles. Participants performed three sets of 12–15 repetitions, with 60–90 s of rest between sets and exercises, totaling approximately 40 min per session. For exercises involving dumbbells or kettlebells, the loads were self-selected based on each participant’s prior experience. During training, participants were instructed and encouraged to perform the exercises at maximal voluntary speed.

In both training modalities, exercises were selected to target muscle groups commonly used in the daily activities of older adults [25]. All sessions were supervised by professionals experienced in exercise prescription for this population. Load testing and training sessions were conducted at the same time of day to ensure consistency.

### 2.5. Data Analysis

Descriptive statistical techniques, including mean and standard deviation, were used to present the variables that characterized the sample. The assumption of data normality was assessed using the Kolmogorov–Smirnov test. A paired Student’s *t*-test was applied to analyze FS and FAS responses across different time points. Spearman’s rank correlation test was used to assess agreement among the data obtained from the scales. Cronbach’s alpha (α) was calculated to estimate the reliability of the questionnaires administered.

Data from the FS, FAS, and Borg scales (during and post-training) were analyzed using IBM SPSS Statistics, version 25. FS scores were converted from the original scale into a fully positive numeric range (1 to 11), thus removing any negative or null values that could interfere with data interpretation.

To determine the strength of the correlations, the following criteria were used: very strong correlation (r ≥ 0.90), strong (0.60 ≤ r < 0.90), moderate (0.30 ≤ r < 0.60), and weak (r < 0.30) [26]. Statistical significance was set at *p* < 0.05.

During the preparation of this manuscript, the author used ChatGPT (GPT-5, OpenAI, San Francisco, CA, USA) to support translation and language editing. The authors reviewed and revised all generated content and take full responsibility for the final manuscript.

## 3. Results

Table 1 presents the characterization of the subjects included in the sample, with pre-training assessments from the two exercise sessions for both groups. As observed, both groups were predominantly composed of women and exhibited cognitive capacity consistent with the exclusion criteria. Furthermore, no significant differences were found in the scale responses during the pre-training assessments of the two exercise sessions.

The descriptive data of the FS, FAS, and BORG scale responses, shown in Table 2, indicate similar responses between the two training sessions, with a slight increase in FS and FAS scores post-training, while perceived exertion (BORG) remained relatively stable. The low standard deviations suggest little variability in the responses.

Table 3 presents the Spearman correlation results regarding the consistency pattern of the scales during and post-training. As observed, both scales demonstrated high consistency, indicating a strong correlation between measurements taken at different time points, which suggests good stability of the measures over time.

Figure 2 presents the reliability coefficients (Spearman’s rho) of the FS, FAS, and BORG scales during and post-training, separated by sex. The results revealed high temporal consistency of the measures across both time points for males and females. Specifically, the FS and FAS scales demonstrated strong reliability during exercise (FS: ρ = 1.000 for males; ρ = 0.955 for females/FAS: ρ = 0.902 for males; ρ = 0.973 for females) and in the post-training phase (FS: ρ = 1.000 for males; ρ = 0.875 for females/FAS: ρ = 0.873 for males; ρ = 0.932 for females). The BORG scale showed moderate yet acceptable correlations (post-training: ρ = 0.775 for males; ρ = 0.892 for females). The standard errors were low across all conditions, with a slight increase in the FAS among females during exercise (SE = 1.125). These findings reinforce the good stability and reliability of the scales when applied at different moments of training and across sexes.

The internal consistency analysis, evaluated by Cronbach’s alpha, showed satisfactory values for the scales used. The FS presented a coefficient of 0.887 for its 11 items, while the FAS yielded 0.860 for 6 items. These values demonstrate good reliability of the scales, indicating that the items consistently measure the intended constructs. Additionally, the intraclass correlation coefficients (ICC) supported these findings, with the FS showing an ICC of 0.925 and the FAS an ICC of 0.869, indicating strong reliability and temporal stability across measurement moments.

## 4. Discussion

The FS and FAS scales have been widely used in exercise research because they allow a direct assessment of participants’ subjective experience, which is important for understanding how individuals feel during physical exercise [27]. Despite these assertions, the reliability of these scales had not yet been assessed in the elderly population. Thus, the aim of this study was to evaluate the reliability of applying the FS and FAS scales in active older adults during resistance training (RT) and functional training (FT).

It is noteworthy that the Spearman correlation coefficients for all comparisons were significant, indicating strong consistency between measurements during and post-training for the FS, FAS, and Borg scales [28]. Furthermore, the correlations of the FS and FAS scales between the two training sessions were particularly high, suggesting excellent internal consistency. Although lower, the correlation of the Borg scale between the two collection moments also indicated good consistency, demonstrating that training intensity was similar between sessions [29]. In addition, Cronbach’s alpha values indicated excellent consistency among the items.

This finding is especially relevant as it ensures that the application of the FS and FAS scales occurred within the same context, given that exercise intensity can influence affective responses, as demonstrated by Ekkekakis and Brand [27]. Moreover, the post-training correlations between visits were also high for all scales, confirming the stability of measurements over time and indicating no significant differences between scales, which reinforces the consistency of their measurements.

In addition to the Spearman correlation coefficients, intraclass correlation coefficient (ICC) values were calculated to further assess the reliability of the FS, FAS, and Borg scales. The ICC is a widely accepted statistical method for measuring the reproducibility of quantitative scores, particularly in test–retest designs [30]. In this study, the ICC values confirmed the excellent reliability of the FS and FAS scales across sessions, with values above 0.90 for both scales in most comparisons. These results reinforce the temporal stability of affective responses in older adults during exercise and support the appropriateness of these scales for repeated use in longitudinal or intervention-based studies. The high ICC values also align with the low standard errors observed, indicating strong measurement precision.

The results presented in this study reinforce the robustness of the FS and FAS scales in assessing affective responses in older adults during physical training. It should be noted that no studies were found that investigated the reliability of these scales for this population; however, Brito and Teixeira [31], when assessing the validation and translation of the FS and FAS scales in Portuguese adolescents, obtained results that corroborate the findings of the present study. Similarly, Thorenz et al. [32] reported positive results when validating the scales in the German version with young adult participants, further supporting the reliability of these scales.

The absence of previous studies assessing the reliability of these scales in the elderly population highlights the relevance of this study, since cross-cultural adaptation and reliability testing of questionnaires are essential to ensure that the instruments used are appropriate for different cultural and linguistic contexts [33]. In this study, the application of the FS and FAS scales in a sample of active older adults allowed for evaluating their effectiveness and confirming their applicability to this population.

Importantly, the practical use of the FS and FAS scales in older adult populations demonstrates considerable potential in both community and clinical settings. Due to their simplicity, low application time, and minimal cognitive demand, these scales are especially suitable for elderly individuals, including those with reduced literacy or mild cognitive decline [31,34]. In community-based physical activity programs, the FS and FAS can serve as accessible tools for monitoring emotional well-being during exercise sessions, supporting the identification of affective patterns that may influence adherence [35]. Likewise, in clinical contexts, these instruments allow health professionals to assess affective responses in real time, facilitating early detection of psychological distress or reduced exercise tolerance [36]. Their ease of administration also enables integration into routine assessments without disrupting therapeutic interventions, reinforcing their value in promoting personalized and emotionally responsive care strategies for aging populations [23].

These findings are particularly important because affective responses to physical exercise are individual-sensitive variables and can significantly impact adherence to training [1]. The ability to accurately measure affect and emotional arousal during exercise contributes to the development of interventions that promote well-being and motivation among older adults [9]. Therefore, the results presented here have significant implications both for professional interventions involving physical exercise and for research.

For health professionals working with physical exercise, the use of the FS and FAS scales offers a valuable tool to monitor individuals’ emotional responses during exercise practice, allowing adjustments in interventions to maximize participants’ pleasure and motivation [9,10]. This is important to promote adherence to exercise programs, as studies such as those by Ribeiro et al. [37] and Ferretti [38] indicate that the maintenance of older adults in exercise programs depends on the pleasure generated by these activities. For the elderly, this is even more relevant [38], as only regular practice favors the development of physical capacities such as muscular strength and cardiorespiratory fitness [39], which are fundamental for maintaining health and quality of life in advanced ages [38].

Furthermore, for research, the reliability of these scales in older adults expands the possibilities for future studies exploring affective responses to physical exercise in different contexts and populations. This ensures that the obtained data are representative and accurate, contributing to a deeper understanding of the interactions between physical exercise and emotional well-being [40].

However, it is important to highlight some limitations of this study. First, it was conducted with a specific sample of healthy, physically active older adults, which may limit the generalizability of the findings to populations with different health conditions. Additionally, emotional responses may have been influenced by uncontrolled contextual variables, such as the training environment and interaction with instructors [40]. Moreover, the study did not evaluate convergent or discriminant validity using other established affective measures, such as the PANAS or POMS. This limitation is justified by the specific design of the study, which aimed to assess the test–retest reliability of the FS and FAS, rather than their construct validity. Nonetheless, future studies should consider more heterogeneous samples, incorporate comparisons with additional affective instruments, and apply the scales across different exercise modalities and practice environments. The inclusion of objective measures, such as heart rate and oxygen consumption, may also be valuable to complement subjective assessments and provide a broader understanding of affective responses to physical activity.

## 5. Conclusions

This study demonstrated the reliability of the FS and FAS scales to assess affective responses in physically active older adults during physical training. The high correlation coefficients and the absence of significant differences between measurements taken during and post-training confirm the robustness of these instruments.

## Figures and Tables

**Figure 1 ijerph-22-01317-f001:**
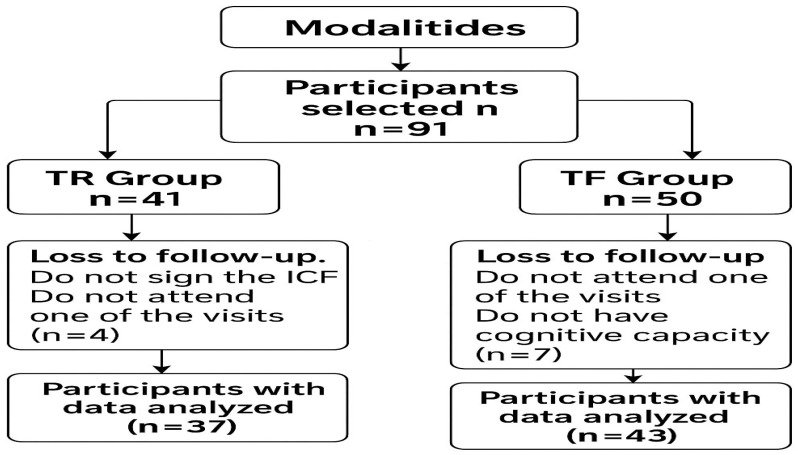
Flowchart—Participant Selection Process.

**Figure 2 ijerph-22-01317-f002:**
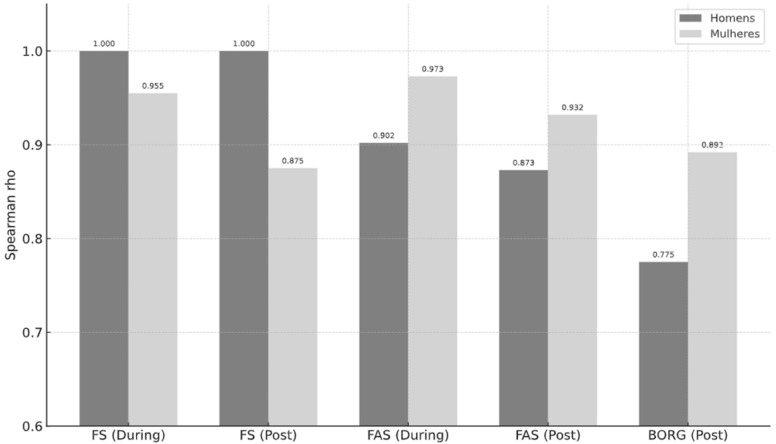
Spearman Correlation Coefficients of FS, FAS, and BORG Scales During and Post-Training by Sex. Legend: FS = Feeling Scale; FAS = Felt Arousal Scale; Post = post-training response; Men = data from male participants; Women = data from female participants; ρ = Spearman’s correlation.

**Table 1 ijerph-22-01317-t001:** Sample characteristics (*n* = 80).

Sample					
Modality	*n*/%	Men (*n*/%)	Women (*n*/%)	Age (Years)	Mini Mental
TR	37/46.3	5/13.5	32/86.5	68 ± 7	28.29
TF	43/53.7	7/16.3	36/83.7	71.5 ± 7.5	28.69
**Assessment of scales at pre-training**					
	**Session 1**		**Session 2**		***p*-Value**
FS	4.81		4.82		0.908
FAS	5.69		5.71		0.819

Legend: TR = Resistance Training; TF = Functional Training; FS = Feeling Scale; FAS = Felt Arousal Scale; *n* = participant.

**Table 2 ijerph-22-01317-t002:** Descriptive data of the scale responses (*n* = 80).

SESSION 1						
	During Training			Post-Training		
	FS	FAS	BORG	FS Post	FAS Post	BORG Post
Mean	4.36	5.23	6.08	4.81	5.69	6.28
DP	1.105	1.125	0.632	0.618	0.704	0.573
**SESSION 2**						
	**During Training**			**Post-Training**		
	**FS**	**FAS**	**BORG**	**FS Post**	**FAS Post**	**BORG Post**
Mean	4.36	5.29	6.06	4.81	5.71	6.28
DP	1.070	1.058	0.581	0.591	0.679	0.551

Legend: *n* = participant; FS = Feeling Scale; FAS = Felt Arousal Scale; Post = post-training response; DP = Standard Deviation.

**Table 3 ijerph-22-01317-t003:** Consistency Results of the FS, FAS, and Borg Scales Between During and Post-Training Moments.

	*N*	Rho	*p*-Value (*t*-test)	Standard Error	DP
FS INT 1–FS INT 2	80	0.936 **	1.000	0.036	0.318
FAS INT 1–FAS INT 2	80	0.967 **	0.058	0.033	0.291
BORG INT 1–BORG INT 2	80	0.786 **	0.783	0.045	0.405
FS POST 1–FS POST 2	80	0.840 **	0.708	0.033	0.297
FAS POST 1–FAS POST 2	80	0.887 **	0.418	0.031	0.274
BORG POST 1–BORG POST 2	80	0.875 **	1.000	0.031	0.276

Legend: FS = Feeling Scale; FAS = Felt Arousal Scale; INT = response during training; 1 = first visit; 2 = second visit; POST = post-training response; DP = Standard Deviation; *n* = participant; rho = Spearman’s correlation; ** = *p* < 0.01.

## Data Availability

The datasets generated and analyzed during the current study are available in the Zenodo repository, at https://doi.org/10.5281/zenodo.16930149 (accessed on 5 March 2025).

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
