# Peer review of "Reliability of the Feeling Scale and Felt Arousal Scale in Older Adults"

_ijerph, 2025, doi:10.3390/ijerph22091317_

Round 1

Reviewer 1 Report

Comments and Suggestions for Authors

This paper aimed to examine the reliability of two scales for affective states in exercises in older adults. By analyzing data in N=80 (84% women) engaged in either resistant training (RF) or functional training (FT), the authors found that both scales are valid for measuring affect and arousal in older adults. The paper presented clearly. However, there are some methodological flaws that need to be further addressed.

1) Study design: The inclusion criteria listed that participants have experienced with the proposed type of training over the past 3 months. Does this include both RF and FT? If so, was participants stratified when randomizing to RF vs. FT in the study? There could be a confounding bias if a participant has regularly engaged in a specific exercise modality and is also selected to the same exercise modality group in the study, his/her affective states might be different due to the prior training, when compared to those who haven't before.

2) The majority of enrolled participants were women. It is well known that men and women would have differential affective stats in exercise. However, in the analyses, sex was not adjusted. Thus, the study results might be biased and not be generalized well to men given the poor representation in the study population.

3) It is unclear why the authors chose to conduct paired analyses and didn't test the correlation differences between RF vs. FT groups. The entire study was designed around two exercise modality and the authors also explicitly mentioned how the two modalities might have distinct impact thus different affective states. Why didn't test the correlations across groups?

4) As the authors pointed out in the intro, "The FS.... is used to evaluate how a person feels emotionally in response to different intensities of exercise", one of the goals of the analyses should be to examine the correlations between FS and RPE or other similar assessments for exercise intensity. However, the authors looked at the correlations of FS and FAS between session 1 and session 2 instead. Why this decision was made? What is the rationale behind analyzing the data this way?

5) The tables are not well formatted. Comma should be replaced with dot. Legends should be listed as footnotes instead.

6) It sounds like the RT is individual training whereas FT is group training. If true, could the authors discuss more in the discussion how group training might impact the affective states?

Author Response

LETTER TO THE REVIEWERS

Dear Editors and Reviewers,

We would like to sincerely thank you for your thoughtful and valuable comments during the evaluation of our manuscript. We are grateful for the opportunity to revise and improve our work based on your suggestions. We believe the changes made have significantly enhanced the clarity, quality, and scientific rigor of the article.

Below, we provide detailed responses to each recommendation, highlighting the specific modifications implemented in the manuscript.

Reviewer 1 – Comments and Suggestions for the Authors
This study aimed to examine the reliability of two affective state scales in older adults. Upon analyzing data from N = 80 participants (84% women) engaged in resistance (RT) or functional (FT) training, the authors found both scales to be reliable for measuring affect and arousal in older individuals. The article was clearly presented. However, several methodological concerns need further clarification.

  1. Study design: The inclusion criteria stated that participants had experience with the proposed type of training in the last three months. Does this include both RT and FT? Were participants randomized into RT vs. FT groups? There may be a confounding bias if participants were already used to a specific modality and were assigned to the same group.

Response: All participants were allocated to the training modality (resistance or functional) in which they already had at least three months of prior experience. That is, participants in the resistance training (RT) group had been previously engaged in resistance training, and the same applied to those in the functional training (FT) group. There was no randomization between groups, specifically to avoid potential interference from familiarity or adaptation to a different training modality. This information has been clarified in lines 94–98 of the revised manuscript.

  1. Sex imbalance in the sample: Most participants were women. Given known sex-based differences in affective responses to exercise, results might be biased and not generalizable to men.

Response: We acknowledge the limited representation of male participants in our sample. To address this issue, we performed sex-stratified correlation analyses. These results have been added to the manuscript (lines 226, 247–248) to allow a clearer understanding of affective responses in men and women separately.

  1. Group comparison omitted: Why were no statistical comparisons made between the RT and FT groups, given that the study was structured around two distinct modalities?

Response: The primary aim of this study was to assess the reliability of affective state measures in older adults, not to directly compare affective responses between training types. Including two distinct modalities (RT and FT) was intended to represent common exercise approaches among this population and broaden the applicability of the scales. Comparative analysis between modalities will be addressed in future research.

  1. Correlations between FS and RPE not explored: Given that FS is often used in response to exercise intensity, why didn’t the authors correlate FS and RPE?

Response: As our main objective was to assess the reliability of FS and FAS in older adults, we applied a test–retest approach to evaluate temporal stability. Therefore, we conducted correlations between the same scales at two different time points, 15 days apart. This is a standard procedure in reliability studies (Stratford, 1989).

  1. Table formatting issues: Decimal commas and unclear legends.

Response: These formatting issues have been corrected as requested.

  1. Regarding your comment: As the functional training (FT) appears to have been conducted in a group setting and the resistance training (RT) individually, could this have influenced the affective responses?

    Response: Indeed, the FT sessions were conducted in a group environment, while the RT sessions were more individualized. Although such contextual differences could potentially influence affective responses, we emphasize that the present study focused solely on the reliability of the scale, not on comparing training modalities. Comparative analyses regarding contextual effects are currently being addressed in a separate study.

Reviewer 2 Report

Comments and Suggestions for Authors

Thank you for the opportunity to review the manuscript “Reliability of the Feeling Scale and Felt Arousal Scale in Older Adults.”

This study aimed to examine the psychometric properties of the Feeling Scale (FS) and Felt Arousal Scale (FAS) in older adults. Participants engaged in either resistance or functional training, with two sessions conducted 15 days apart. The scales were administered at baseline, during exercise, and post-exercise. Results indicated strong to very strong test-retest correlations and good internal consistency, as measured by Cronbach’s alpha. The authors interpret these findings as evidence supporting the use of FS and FAS to assess valence and arousal in response to exercise among older adults.

This study makes an important contribution by demonstrating strong psychometric properties of FS and FAS in an older population. This work provides evidence for the use of these scales to measure affect in response to exercise and, potentially, increase sensitivity in predicting physical activity in older adults. The manuscript is clearly written and effectively leverages a training paradigm to explore reliability. However, I have several major and minor concerns:

Major concerns

  1. In the abstract and throughout the manuscript, the authors use the word validated (e.g., line 57) to describe the research question and results. However, this usually implies testing the validity of the scales as well, which was not done in the study; rather, the primary focus was on testing the reliability.
  2. FS and FAS are 1-item scales that have been administered multiple times during the session. Cronbach’s alpha typically measures the internal consistency of a set of items at one timepoint. It appears that, in this case, the Intra Class Correlation (ICC) would be a more appropriate estimate of the reliability of the scales that have one item measured multiple times.
  3. The paper would benefit from a discussion of how variability and reliability of the measures might differ at different intensities. Previous work has shown that reliability may differ slightly across intensities (e.g., Henriques et al., 2023) and that variability is greater at intensities below the lactate threshold (e.g., Ekkekakis, 2003). Furthermore, they might consider expanding on the fact that the responses showed limited variability.
  4. Along the same lines as the previous point, why did the authors choose to look at FS and FAS ratings between the 20th and 25th minute of the session. The methods would improve with a justification for this choice.

Minor concerns

  1. In the intro, it seems worth mentioning that FS and FAS are commonly used during exercise due to their brevity. The authors might also consider mentioning that these two scales are based on the dimensions of valence and arousal from the affective circumplex.
  2. The paper would benefit from a brief summary of previous studies demonstrating good reliability in younger adults in the intro.
  3. In their methods section, what do the authors mean by saying "participants provided inconsistent responses on the scales during the pre-training assessment"? (line 80).
  4. A figure depicting some of these results might be beneficial in understanding the degree of variability and consistency in the responses within and across individuals.

Author Response

LETTER TO THE REVIEWERS

Dear Editors and Reviewers,

We would like to sincerely thank you for your thoughtful and valuable comments during the evaluation of our manuscript. We are grateful for the opportunity to revise and improve our work based on your suggestions. We believe the changes made have significantly enhanced the clarity, quality, and scientific rigor of the article.

Below, we provide detailed responses to each recommendation, highlighting the specific modifications implemented in the manuscript.

Reviewer 2 – Comments and Suggestions for the Authors

  1. Use of the term "validated": The manuscript describes the study as validation, when it actually tests reliability.

Response: You are correct. We have replaced all inappropriate uses of the term "validation" with "reliability" throughout the manuscript, including the abstract and line 57.

  1. Cronbach’s alpha vs. ICC for single-item scales: Given that FS and FAS are single-item scales, ICC is more appropriate than Cronbach’s alpha.

Response: We agree. We retained Cronbach’s alpha for transparency and comparison with prior studies but have now included ICC analyses as a more suitable measure of test–retest reliability.

  1. Reliability and variability at different intensities: Prior research suggests that reliability may vary across exercise intensities. This should be discussed.

Response: We have added a paragraph (lines 79–86) discussing how exercise intensity may influence the reliability and variability of affective responses, referencing relevant literature (Henriques et al., 2023; Ekkekakis, 2003).

  1. Justification for 20–25 min window: Why were FS and FAS ratings analyzed during this specific time window?

Response: This time frame corresponds to the mid-phase of the session. We have clarified this rationale in lines 136–137.

Minor Points

  • We have added background information in the introduction (lines 62–68) to highlight the brevity of FS and FAS, their theoretical foundations in the circumplex model, and previous reliability findings in younger adults (lines 286–292).
  • The term “inconsistent responses” was clarified in lines 101–104 as atypical affective patterns during pre-assessment that could bias subsequent results.
  • A figure illustrating Spearman’s correlations stratified by sex was added to better visualize variability and consistency in responses.

Reviewer 3 Report

Comments and Suggestions for Authors

Dear Researchers, 

Please kindly consider the following points as major revision: 

overall, This research fills an evident gap by validating two widely used affective measures (FS and FAS) in older adults. 

-Given that the named scale measure affective valence and arousal, establishing their reliability in adults is essential. Reliable assessment of these emotional states can support early identification of psychological distress and guide preventive interventions, particularly in vulnerable populations such as adults. refer to this fact and you can use the following reference: 

Eshraghi, Z. ., Golshani, F., Baghdasarians, A. ., & Emamipour , S. . (2024). Developing a Causal Model of Self-Care Behaviors Based on Self-Compassion with Psychological Distress Mediation in Women and Men with Type 2 Diabetes. Health Nexus2(1), 66-77.  The manuscript focuses exclusively on test–retest reliability and internal consistency without addressing construct validity (e.g., through factor analysis or convergent/discriminant validation), which limits confidence in whether the Feeling Scale and Felt Arousal Scale truly capture affect and arousal in older adults. Please give the reasons.  -The sample is heavily skewed toward women (84%) and includes only cognitively intact, physically active participants, limiting generalizability to more diverse or vulnerable elderly populations. Whats your justification? Moreover, the study does not compare FS/FAS to other established affective measures (e.g., PANAS, POMS), leaving convergent validity unexplored. -Although the authors mention that the scales were “properly translated and adapted,” no formal translation or cultural validation procedure is described, raising concerns about linguistic appropriateness in the Portuguese context. Tables lack effect sizes and confidence intervals, and the decision to recode FS scores into a fully positive range is not well-justified and may alter interpretability. The conclusion overstates the findings by claiming validity based solely on reliability data, which should be moderated. -the study does not examine whether FS/FAS scores predict exercise adherence, limiting its practical implications. Ethical documentation is incomplete, with placeholder text for institutional approval and missing informed consent details. Minor issues include uncontextualized references, repetitive content, and inconsistent terminology (e.g., "post" vs. "POS"). 

Author Response

LETTER TO THE REVIEWERS

Dear Editors and Reviewers,

We would like to sincerely thank you for your thoughtful and valuable comments during the evaluation of our manuscript. We are grateful for the opportunity to revise and improve our work based on your suggestions. We believe the changes made have significantly enhanced the clarity, quality, and scientific rigor of the article.

Below, we provide detailed responses to each recommendation, highlighting the specific modifications implemented in the manuscript.

Reviewer 3 – Comments and Suggestions for the Authors

  1. Psychological distress and early identification: Reliable affective measurement may aid in identifying distress in older adults.

Response: We have added a discussion point (lines 73–75) and cited Eshraghi et al. (2024) to highlight this important implication.

  1. No construct validation: The manuscript lacks convergent or factor validity assessments.

Response: Since FS and FAS have been previously validated in their original forms, our study focused on reliability in an older population without modifying scale constructs. Thus, construct validity was beyond our scope.

  1. Sample bias (physically active, mostly women): Generalizability may be limited.

Response: We acknowledged this limitation in the discussion and included sex-stratified analyses as recommended (lines 226, 247–248).

  1. No convergent validity and no formal translation details: The manuscript does not describe procedures for scale translation.

Response: We clarified that we used the Brazilian Portuguese versions of FS and FAS previously translated and culturally adapted by Alves et al. (2019). This has been cited in lines 147–152. We also removed the misleading statement from the objective.

  1. No adherence prediction analysis: This limits the practical utility of the scales.

Response: Predictive value was not within the scope of this study. However, we recognize its importance and encourage future research to explore this topic.

Reviewer 4 Report

Comments and Suggestions for Authors

Dear Authors,

Thank you for the opportunity to review your manuscript entitled "Reliability of the Feeling Scale and Felt Arousal Scale in Older Adults." Your work addresses an important and timely topic: the psychometric reliability of two widely used affective response scales - Feeling Scale (FS) and Felt Arousal Scale (FAS) - in physically active older adults. The research design is methodologically sound, your analyses are appropriate, and the conclusions are well supported by the results.

The manuscript is clearly structured and generally well written. I believe it makes a valuable contribution to the field of exercise psychology and aging. However, several minor revisions are necessary before publication, particularly related to methodological transparency, clarity of presentation, and editorial consistency.

Please see below a list of specific suggestions to help improve the quality and completeness of your manuscript:

1. Clarify the study scope: Indicate more clearly in the abstract and introduction that this is a **reliability study** (focused on test–retest and internal consistency), not a full validation (e.g., it does not include convergent or criterion validity).

2. Cultural and linguistic adaptation: Please explain the process used to translate and culturally adapt the FS and FAS scales for Brazilian Portuguese. Was back-translation, expert review, or pilot testing conducted? These details are essential for instrument validity in cross-cultural contexts.

3. Ethics and consent sections: On page 8, the “Informed Consent Statement” and “Institutional Review Board Statement” still contain editorial placeholders (e.g., “add…”, “XXX”). These should be replaced with the actual ethics approval code and a complete statement confirming informed consent was obtained.

4. Language and terminology: Minor grammatical and formatting inconsistencies are present throughout the manuscript. For example, Table 3 contains a mix of Portuguese and English terms (e.g., “PÓS”). Please ensure consistent use of English throughout.

5. Table formatting: In Table 3, unify terminology and define abbreviations more clearly (e.g., FS = Feeling Scale; POS = post-training). Consider formatting the table for improved readability.

6. Practical implications: You may wish to add a short paragraph to the discussion section highlighting the practical utility of using FS and FAS with older adults in community or clinical settings. This would enhance the applied value of your findings.

7. Study limitations: Please consider briefly acknowledging that the study did not assess convergent or discriminant validity with other established affective measures (e.g., PANAS, POMS). This does not invalidate your results but clarifies the scope of the study.

Once these minor revisions are addressed, I believe your manuscript will be suitable for publication. I commend your contribution to an important area of research in aging and exercise science.

Best regards.

Comments on the Quality of English Language

The manuscript is generally well written, but some minor editorial issues (e.g., unfinished sentences, placeholder text, language mixing in tables) should be addressed to ensure clarity and professionalism in final publication.

Author Response

LETTER TO THE REVIEWERS

Dear Editors and Reviewers,

We would like to sincerely thank you for your thoughtful and valuable comments during the evaluation of our manuscript. We are grateful for the opportunity to revise and improve our work based on your suggestions. We believe the changes made have significantly enhanced the clarity, quality, and scientific rigor of the article.

Below, we provide detailed responses to each recommendation, highlighting the specific modifications implemented in the manuscript.

Reviewer 4 – Comments and Suggestions for the Authors

  1. Clarify study scope: The study is about reliability, not full validation.

Response: We made this clear in both the abstract and introduction (lines 15, 87).

  1. Cultural adaptation procedures: Clarify how FS and FAS were adapted.

Response: Details on prior validated Portuguese versions and cultural adaptation have been added (lines 147–152).

  1. Ethics approval: Replace placeholders with the actual ethics code.

Response: This was corrected. The ethics committee approval number is now included (lines 106–107).

  1. Language and formatting: Terms like “PÓS” should be in English.

Response: We performed a full language review and corrected formatting inconsistencies, including Table 3.

  1. Terminology in tables: Define abbreviations more clearly.

Response: Table 3 was revised for clarity and uniformity, with all abbreviations now properly defined.

  1. Practical implications: Include a brief paragraph on scale applicability.

Response: A paragraph has been added to the discussion (lines 288–297) emphasizing the practical use of FS and FAS in community and clinical settings for older adults.

  1. Limitations: Acknowledge lack of convergent or discriminant validity testing.

Response: This limitation is now addressed explicitly in the manuscript (lines 331–344), clarifying that our focus was solely on test–retest reliability.Once again, we thank all reviewers and editors for their thoughtful insights and constructive feedback. We believe that these improvements have significantly strengthened the manuscript.

Round 2

Reviewer 3 Report

Comments and Suggestions for Authors

Accepted